# Host Plants for the Lanternfly *Saiva formosana* Kato, 1929 (Hemiptera, Fulgoridae) Endemic to Taiwan, and Parasitism of Its Eggs by Wasps

**DOI:** 10.3390/insects15110841

**Published:** 2024-10-26

**Authors:** Meng-Hao Hsu, Meng-Ling Wu, Liang-Jong Wang

**Affiliations:** 1Taimalee Research Center, Taiwan Forestry Research Institute, Ministry of Agriculture, Taitung 963001, Taiwan; catflea@tfri.gov.tw; 2Taiwan Forestry Research Institute, No. 67, Sanyuan St., Zhongzheng Dist., Taipei City 10079, Taiwan; mlw@tfri.gov.tw

**Keywords:** egg mass, *Pyrops candelaria*, *Pyrops watanabei*, *Ficus fistulosa*, *Elaeocarpus decipiens*, *Heptapleurum heptaphyllum*

## Abstract

*Saiva formosana* Kato, 1929 is a species of lanternfly endemic to Taiwan. To date, there have been few reports on the morphology of adults and the fifth-instar nymph of this species. *Saiva formosana* can be confused with *Pyrops watanabei* (Matsumura, 1913) due to the morphological similarities exhibited by the nymphs of these two species. *Elaeocarpus decipiens* F. B. Forbes & Hemsl. has recently been reported to be the sole host plant for *Saiva formosana*. However, hatched egg masses of *Saiva formosana* have been detected on many other plant species. For the present study, we undertook a one-year investigation, lasting from May 2023 to April 2024, to record the different life stages of *Saiva formosana* and identify host plants. During the investigation, some egg masses were removed from tree trunks and reared in the laboratory in order to verify the species and explore any possible interaction between the hatching of this lanternfly species and the emergence of parasitic wasps. Finally, we compared the oviposition strategies and parasitic behaviors of wasps among the three lanternfly species, namely *Saiva formosana*, *Pyrops watanabei*, and *Pyrops candelaria* (Linnaeus, 1758).

## 1. Introduction

Invasive lanternflies, such as the notorious *Lycorma delicatula* (White, 1845), pose a severe threat to agriculture and the broader ecological environment [1]. In Taiwan, the longan lanternfly *Pyrops candelaria* (Linnaeus, 1758) has been reported since 2018 [2]. Although the impact on agriculture of this introduced lanternfly is not that serious compared to *L. delicatula*, its influence on the fauna and flora is still worth studying [3,4]. It may be the case that closely related native species are most vulnerable to invasion by *P. candelaria* due to their similar habitats. The more knowledge we obtain concerning these species, the clearer picture of their interaction we will gain. Ferrenberg and Denno [5] indicated that there has been a difference in opinion over the importance of competition, host plant resources, and natural enemies in community structures. Recent reviews have shown that interspecific competition can be an important factor in affecting the performance and population dynamics of herbivorous insects. In recent studies, the relationships between host plants and different stages of *P. watanabei* (Matsumura, 1913) have been established following the introduction of *P. candelaria* [2,6]. The introduced host plant, *Triadica sebifera* (L.) Small belonging to Euphorbiaceae, is relevant to the developmental stages of both lanternfly species. The coexistence of these two lanternfly species may complicate control strategies for *P. candelaria* [2,6]. In addition, during previous investigations, a third species of lanternfly, *Saiva formosana* Kato, 1929, was detected in some communal habitats.

A total of 13 species of *Saiva* are known to be distributed in Asian countries such as Cambodia, India, Indonesia, Malaysia, Myanmar, Sri Lanka, and Thailand [7]. In Taiwan, however, only *S. formosana* is considered an endemic species [7]. *Saiva* is one of the genera most closely related to *Pyrops* [8,9], and adults are easily differentiated, but eggs and nymphs are more difficult. After the discovery of *S. formosana* in 1929 [10], there were almost no further reports on this species until 2023, when *Elaeocarpus decipiens* F. B. Forbes & Hemsl. was reported to be the host plant for adults and only one fifth-instar nymph of this species [11]. The scarcity of references to *S. formosana* in the literature means that more information is required on the fundamental characteristics of this species. However, as we developed the ability to identify egg masses of *S. formosana*, we were able to determine that this species has a wide range of distribution in the mountainous areas of Taiwan. The findings of host plant shift obtained for the nymphal and adult stages of two *Pyrops* species suggested that *S. formosana* is probably hosted by plants other than *E. decipiens* at different developmental stages and times of the year [2,6].

In previous research, we also observed that egg parasitism by wasps upon *S. formosana* is prevalent. Therefore, in the present study, we sought to determine the range of hosts across different developmental stages, the preferred egg-laying sites, and the incidence of parasitism by wasps upon egg masses. In addition, we explored the relationship between the parasitism rate and the survival rate for the eggs of this lanternfly. In the United States and South Korea, an egg parasitoid, *Anastatus orientalis* Yang and Choi, 2015, has been selected and evaluated as a biological control agent for managing exotic *Lycorma delicatula* [12]. In the case of *Pyrops candelaria*, no implication of egg parasitoids has yet been taken as the control agent. However, an ancient record of the emergence holes of wasps on egg masses has been published [13]. Further research is required to determine whether the egg-parasitic wasp frequently found in the present study has the potential for expanding its host range from *S. formosana* to *P. candelaria*, either as a result of human activities or through gradual natural processes. In the Discussion Section of this article, we compare the oviposition strategies and parasitic behaviors of wasps among three species of lanternfly, namely *S. formosana*, *P. watanabei*, and *P. candelaria*.

## 2. Materials and Methods

Prior to this study, we detected many hatched egg masses that had no traces of wax and, therefore, were unlike the egg masses of *Pyrops watanabei* (Matsumura) or *Pyrops candelaria* (Linnaeus). We suspected that those egg masses belonged to *Saiva formosana* Kato. Accordingly, for the present study, we chose investigation sites (Table 1) with abundances of these hatched egg masses, located in Taipei City, New Taipei City, and Ilan County. We then investigated the different stages of *S. formosana* and their occurrences on various plants over the course of a single year from 1 May 2023 to 30 April 2024.

The investigations were conducted at least twice every week in the daytime, and the first individual detected on 1 May 2023 was an adult on *Elaeocarpus decipiens* F. B. Forbes & Hemsl. The numbers of adults on each plant were noted until the last adult was found on 22 September of the same year. The plant species were also recorded when individuals were detected. In line with the recent studies on the genus *Pyrops*, the scientific names of the species and families of the plants used in this study referred to the same source [2,6]. The full scientific names with authors and date of publication are listed in Appendix A. Moreover, to avoid confusion with egg masses laid by previous generations (i.e., prior to 2023), only unhatched masses were recorded in the present study. The egg masses of *S. formosana* are exposed without any cover, so it is easy to count them using photographs. The number of columns and of eggs per egg mass can then be noted. In the present study, we counted eggs from the first column on the left end of the mass and named the columns C1, C2, C3, and so on, moving rightwards, following Liu [14]. In addition, some egg masses were removed and reared in the laboratory. This was to confirm the species and to quickly distinguish between developing instars in the field. The nymphs subsequently reared from these eggs were taken to be the references of five instars, which differed in both size and color; they were used to differentiate developmental instars and then count the numbers of each instar during our investigation. As the side view of *S. formosana* nymph might be taken to be a nymph of *P. watanabei* [15], we took both dorsal and lateral profile photographs of each nymphal instar (Figure 1 and Figure 2). After 22 September 2023, no more adults were found until 20 April 2024, and our one-year study finally ended on 30 April 2024. Occurrences were recorded using visual means, typically the naked eye aided by an MT 14 flashlight (Ledlenser GmbH & Co. KG, Solingen, Germany) in high luminosity (up to 1000 lumens) as well as via a digital camera with a 60× optical zoom (COOLPIX B700, Nikon Co. Ltd., Tokyo, Japan), if we needed to check the higher parts of the canopy and on overcast days. A total of 40 egg masses were removed from the investigation sites and reared in the laboratory to explore the relationship between the survival of eggs and parasitism by their wasps (*Anastatus* sp.). The rearing period lasted from 7 June 2023 to 4 April 2024.

## 3. Results

### 3.1. Diagnosis of the Nymphs

In terms of the first-, second-, and third-instars of *Saiva formosana* Kato, the dorsal view of the abdomen is light color in the middle and darker on both edges. On the contrary, the abdomens of the nymphs of *Pyrops candelaria* and *P. watanabei* in the first to fourth instars are dark color in the middle and light color on both sides (see photographs in Appendix A). The shape of the cephalic process is also a useful trait to distinguish among the three species. From lateral views, the cephalic processes of the fourth- and fifth-instar nymphs of *S. formosana* are flat like a sword; however, those of *Pyrops* spp. are curvy like the nip of a fountain pen (see photographs in Appendix A).

### 3.2. Monthly Records of Different Stages

It can be seen from Figure 3 that in May 2023 and April 2024, fifth-instar nymphs and adults were both detected. In June 2023, 55.2% of the adults were recorded, and newly laid egg masses appeared. In addition, some newly hatched nymphs were found beside the egg masses. In July, the recorded proportion of egg masses reached a peak of 47.4%, while the number of adults declined distinctly. In August, we detected most stages, including adults, egg masses, and nymphs in the first to third instars. The nymphs in the third instar were recorded from August to December. Nymphs in the fourth instar were recorded for almost a full half-year, from October 2023 to March 2024. During the investigation in the autumn and winter, most of the nymphs were found to attach closely to twigs with their ventral side, especially at slightly concave points (Figure 4). Only nymphs in the fourth instar were detected in January and February. Nymphs of the fifth instar began to appear again from March 2024 onwards.

### 3.3. Records of Lanternfly Stages and Their Host Plants

#### 3.3.1. Adults

As we can see in Table 2, only six species of plants hosted the adults of *S. formosana*. Adults were detected from April to September, as shown in Figure 3, and 73.4% of these were found on *Elaeocarpus decipiens* F. B. Forbes & Hemsl. Approximately 14.7% of the adults were recorded on *Magnolia compressa* Maxim., and almost 7.0% were on *Tetradium glabrifolium* (Champ. ex Benth.) T. G. Hartley. On only one occasion were adults recorded on *Triadica sebifera* (L.) Small; however, on this occasion, five individuals appeared simultaneously, accompanied by 20 adults of *Pyrops watanabei*. Figure 5B shows a photograph of four lanternflies taken at that time. No adults were recorded on *Heptapleurum heptaphyllum* (L.) Y. F. Deng, *Ficus fistulosa* Reinw. ex Blume, *Machilus zuihoensis* Hayata, 1911, or *M. thunbergii* Siebold & Zucc. The average was 1.7 adults per plant. Adults feeding on the trunk or branch were sighted on *E. decipiens*, *M. compressa*, and *T. glabrifolium* (see Appendix A).

#### 3.3.2. Egg Masses

Although egg masses can be found on the higher parts of trunks or branches, most of the egg masses were detected on the bases of tree trunks, at heights below 50 cm. Only unhatched egg masses (Figure 6A) without any emergence holes (Figure 6B) were counted. As eggs can be re-covered by lids after hatching (Figure 6C,D) and because in the present study, hatched egg masses sometimes retained all their lids, our experience as researchers was required to avoid confusion between hatched and unhatched egg masses when recording our data. Most of the parasitic wasps were observed on newly laid egg masses (Figure 6A); however, we did observe a parasitic wasp attempting to oviposit on an old, hatched egg mass (Figure 6E). In total, 116 unhatched egg masses with 4291 eggs were found on 107 plants, including five dead tree trunks during the period from June to September 2023 (Table 3). The host plants belonged to 29 species of 21 families, and all of them were woody plants. Approximately 19.0%, 14.7%, and 7.8% of the egg masses were found on *Heptapleurum heptaphyllum*, *Magnolia compressa*, and *Machilus zuihoensis*, respectively. Only six egg masses were recorded on *Elaeocarpus decipiens*. A single egg mass was detected on *Ficus fistulosa*. No egg masses were recorded on any herbaceous plants. A total of 17 wasps (*Anastatus* sp.) were observed searching or ovipositing on 12 of the egg masses. The average number of egg masses per plant was nearly 1.1. Egg masses had between eight and forty-nine eggs, with an average figure (with standard error) of 37.0 ± 0.7. Furthermore, the eggs were arranged in one to seven columns (mean ± SE = 4.8 ± 0.1 columns/egg mass), with each column having between one and thirteen eggs. The average numbers of eggs per column, from the left to the right, were as follows: C1 = 6.6; C2 = 8.9; C3 = 8.8; C4 = 8.4; C5 = 6.8; and C6 = 4.1 eggs. Only one egg mass had a seventh column (C7); two eggs were recorded in this instance.

#### 3.3.3. Nymphs

Nymphs usually remained under branches or twigs. In total, 617 nymphs were found on 477 plants, more or less throughout the year from May 2023 to April 2024 (Table 4). The host plants belonged to 13 species of 12 families. Nearly 32.4% and 30.6% of the nymphs were recorded on *Ficus fistulosa* and *Heptapleurum heptaphyllum*, respectively. Moreover, 20.3% were recorded on two *Machilus* species; namely *M. zuihoensis* and *M. thunbergii*. All the nymphal stages were recorded on *F. fistulosa*, although only one egg mass was detected on this tree, as can be seen in Table 3. Only one nymph was recorded on *Elaeocarpus decipiens*. On *Magnolia compressa*, a batch of 36 newly hatched nymphs were observed beside an egg mass. Excluding newly hatched nymphs close to egg masses, nearly 87.3% of the first- and second-instar nymphs were recorded on *F. fistulosa*. Approximately one-half of the third-instar nymphs were found on *F. fistulosa*, with the other half being found on *H. heptaphyllum* and *M. thunbergia*. More than 41.2% of the fourth-instar nymphs were found on *H. heptaphyllum*. During winter, we made three sightings of the fourth-instar nymphs of this lanternfly, together with the nymphs of *P. watanabei* in the third or fourth instar, on the same tree of *H. heptaphyllum* (Figure 5A). Over 62% of the nymphs in the last instar were found on *H. heptaphyllum*, though we did not detect any adults on this plant. The average was 1.3 nymphs per plant. Feeding by nymphs was difficult to be observed in the field because the proboscis is short and hidden between legs. However, more than 50 nymphs and two instars were detected on *F. fistulosa*, *H. heptaphyllum*, *M. zuihoensis*, and *M. thunbergii*. Therefore, these four species of plants might be food plants and winter shelters for nymphs. Other plants may be served as the egg-laying or temporary resting sites in which the newly hatched or fourth-instar nymphs can be detected.

### 3.4. Changes in Host Plant Preference 

#### 3.4.1. According to Time

As can be seen in Figure 7, a trend in the host preference of nymphs toward *Ficus fistulosa* suddenly emerged, at a peak rate of 90.0%, in August 2023, before declining to approximately 55.4% and 46.2% in September and October, respectively. The preference of nymphs for *F. fistulosa* then continued to decline slowly, from 30.3% to 19.2%, between November 2023 and April 2024. In contrast, another trend was recorded for occurrence on *Heptapleurum heptaphyllum*; this emerged after the end of summer and increased with time from 20.0% in September 2023 to 48.7% in December. The level of occurrence then remained above 30% from January to April 2024. Another trend was recorded for the occurrence of nymphs on *Machilus zuihoensis* and *M. thunbergii*. This ranged from 10% to 25% between September and December of 2023, but then rose to a higher level of between 29% and 46% from January to April 2024. Finally, the trend for the occurrence of nymphs on plants other than the four species mentioned above was remarkably 100% in both June and July 2023, but less than 20% in all the other months.

#### 3.4.2. According to the Developmental Stage

Two opposite trends in host plant preference according to different developmental stages can be seen in Figure 8. First, the preference of nymphs toward *Heptapleurum heptaphyllum* steadily increased from 1.5% in the second instar to 62.2% in the fifth instar. Over the same developmental period, the preference for *Ficus fistulosa* decreased from 86.2% to 13.3%. In addition, the possibility of detecting nymphs on *Machilus zuihoensis* and *M. thunbergia* was approximately 10–20%. Finally, 68.1% of the egg masses and 88.2% of the nymphs in the first instar were recorded on plants other than the four species mentioned above.

### 3.5. Egg Masses and Parasitic Wasps

In total, 1529 eggs were counted from 40 egg masses. In terms of the eggs, the hatching rate was 45.5% (695 first-instar nymphs), and the emergence rate was 20.9% (320 adult wasps). In addition, no more than one adult wasp emerged from any individual egg, and no more than one emergence hole was left above any lid (Figure 6B). The remaining eggs were deemed abortive (514 eggs, 33.6%), as no insects were harvested. In terms of the egg masses (Table 5), more than 40% were with parasitic wasps. Overall, 90% of the egg masses either emerged only adult wasps (lose–win situation) or hatched only nymphs of the lanternfly (win–lose situation).

## 4. Discussion

This is the first article to reveal the morphology of all the developmental stages in a species of the genus *Saiva*, from egg masses, to nymphs in five different instars, to adults. The adults and fifth-instar nymphs of *S. formosana* Kato [11] and *S. gemmata* (Westwood) [16] have previously been described in the literature. These two species can be easily identified by their respective morphologies, and from color patterns visible in the ecological photographs taken in the field [16]. However, to avoid any risk of mistaken identity, photographs of dorsal and lateral profile views should be taken for reference. In previous studies, we obtained photographs of *S. formosana*, *Pyrops watanabei* (Matsumura), and *P. candelaria* (Linnaeus) for the purposes of comparison. In the back and side views, it can be seen that in the first- and second-instar nymphs of these three lanternfly species, the head and thorax are both blacker and darker, compared with the third-, fourth-, and fifth-instar nymphs [6,15]. In light of the findings of the present study, we suggest that the lateral view of the nymph reported in Constant and Pham [15] is a third instar of *S. formosana*, instead of *P. watanabei*. Furthermore, the color of the older three instars, including the third, fourth, and fifth instars, of *P. watanabei* is just brown with light patterns [6,15], whereas those of *S. formosana* are brown with a shade of green (Figure 1D,F, Figure 2C–E and Figure 4; the photograph in Constant and Pham [15]) or red (Figure 1E and Figure 4A). These two lanternfly species could exist in the same habitat, even on the same tree. Were they to coexist, it is most likely that their nymphs in the third or fourth instar would be found on *Heptapleurum heptaphyllum* (L.) Y. F. Deng in the proximity of *Triadica sebifera* (L.) Small during overwintering [6]. In the present study, we detected five adults of *S. formosana* on *T. sebifera*. In the previous studies, it was found that nymphs and adults of *P. candelaria* and *P. watanabei* had preferences towards *T. sebifera* [2,6]. In light of these findings, it is reasonable to assume that the coexistence of the above-mentioned three lanternfly species will become widespread in Taiwan in the future. In addition, the results of the present study indicate that most of the main host plants for *S. formosana* and *P. candelaria* are different. Hence, we may speculate that the greatest survival stress facing *S. formosana* is from egg parasitism rather than invasion by *P. candelaria*. By contrast, it may be that the stress facing *P. watanabei* from invasion by *P. candelaria* is worthy of greater concern, owing to their communal key hosts, *T. sebifera* and *T. cochinchinensis* Lour., for nymphs as well as adults.

The egg masses of *S. formosana* are similar to those of *P. candelaria* and *P. watanabei*, except for three biological traits. First, there are no wax covers on the egg masses of *S. formosana*; in contrast, the egg masses of *P. candelaria* [2] and *P. watanabei* [6,17] have thin and thick layers, respectively, of white wax. The record for the existence of *Lycorma delicatula* (White) in Taiwan is problematic according to Lin et al. [11], but the egg masses of endemic *L. meliae* Kato do have a covering layer of gray wax [Hsu, unpublished data]. Second, the size of the egg mass matters. In this study, we established that *S. formosana* has the lowest number of eggs in masses, with average figures of 37.0 eggs and 4.8 columns per mass. These may be compared with corresponding figures (unpublished data, Hsu) for *P. candelaria* (*n* = 30, mean ± SE = 82.5 ± 3.8 eggs, 8.4 ± 0.3 columns) and *P. watanabei* (*n* = 30, 134.8 ± 2.6 eggs, 11.4 ± 0.4 columns). The egg mass of *Penthiocodes atomaria* (Weber, 1801) is also without a wax cover; however, the size of this mass is bigger than that of *S. formosana* [18]. Finally, the levels of egg parasitism are different in the three species. The prevalent egg parasitism in *S. formosana* leads to the frequent appearance of emergence holes on hatched egg masses. In comparison, in the cases of *P. candelaria* and *P. watanabei*, egg parasitism by wasps was found to be rare in the former, and absent in the latter. No parasitic wasps were observed during our previous investigation of 186 egg masses of *P. watanabei*, and only 2 out of 203 egg masses of *P. candelaria* were found with ovipositing parasitic wasps (unpublished data, Hsu). According to Fatouros et al. [19], egg coating may be considered one of the factors leading to escape from egg parasitism. Regarding *S. formosana*, the exposed egg mass may play an important role in the prevalence of egg parasitism. However, the smaller sizes of egg clutch may be the reason why this species still exhibited a 45.5% egg survival rate in the laboratory in the present study. The result of the egg-mass rearing experiment conducted in this study revealed that after being parasitized by wasps, most of the masses had no eggs that survived. We think a much better hatch rate may be obtained in the field based on the report of *L. delicatula* [20]. Many species in Fulgoridae are characterized by the production of copious amounts of wax [21]. Insect wax may have other functions than escaping from predation and parasitism, such as camouflage [22], protection against desiccation [23], etc. Therefore, the issues of full functions performed by wax coverings on lanternflies need more research to address. Moreover, the types and compositions of waxes in adults, nymphs, and eggs are different [24]. In terms of *S. formosana*, no wax was observed on adults or egg masses, but wax attachments were found on the rear ends of nymphs (Figure 1B,C and Figure 4A). In addition, wax was often seen on the nymphs of *S. formosana, P. watanabei*, and *P. candelaria* during rearing in our laboratory. Bosuang et al. [25] indicated that wax is protective against predators and mildew.

Knowledge of changes in host plants can be crucial when evaluating the ecological impacts of the invasive species and determining control strategies [2]; however, the strategy of changes in host plants may be utilized by female insects to oviposit on a broader range of hosts; such insects may even oviposit on unsuitable hosts to make it harder for predators or parasitoids to find their eggs and early-stage instars [26,27]. In this study, *S. formosana* had a broad range of oviposition sites, including 29 species of plants. Additionally, nymphs preferred host plants that differed from those preferred by adults. The host preference for egg-laying by female *S. formosana* indicated that general site criteria may be more important than species-specific criteria. Like the cases in *P. candelaria* and *P. watanabei*, some egg masses were laid on dead trees [2,6].

For the convenience of investigation, we chose close-at-hand locations with low elevations (118–354 m) in northern Taiwan. However, in June 2024, we found that the distribution of this species could include areas at an elevation as high as 1300 m, in Jinfeng Township, Taitung County, in the mountain of the southern tip of Taiwan (unpublished data, Hsu). The question of whether the preference of this species towards its main host plants is different on higher mountains needs further study to answer. Lin et al. [11] pointed out that *Elaeocarpus decipiens* was the sole host for the adults and a nymph of *S. formosana*. In this study, we recorded the adults of *S. formosana* mostly in May and June, when *E. decipiens* bloomed [28]. Nevertheless, we also recorded more than ten adults on both *Magnolia compressa* Maxim. and *Tetradium glabrifolium* (Champ. ex Benth.) T. G. Hartley. In addition, in the present study, we established that nymphs had no preference towards *Elaeocarpus decipiens*. Instead, we found that nymphs had preferences for *Ficus fistulosa* Reinw. ex Blume, 1825, *H. heptaphyllum*, *Machilus zuihoensis* Hayata, and *M. thunbergii* Siebold & Zucc.

Lanternfly species have diversified life histories. The nymphs of *S. formosana, P. watanabei*, and *P. candelaria* require five instars to become adults, whereas those of *L. delicatula* require only four [29]. Although *Saiva* is the genus most closely related to *Pyrops*, the life history of *S. formosana* is more similar to that of *P. watanabei* than *P. candelaria* due to a longer nymphal stage and the tendency of overwintering on blooming *H. heptaphyllum* [30]. Mainly, nymphs in the fourth instars of *S. formosana* and the third or fourth instar of *P. watanabei* take *H. heptaphyllum* as winter shelters [6], whereas *P. candelaria* and *L. meliae* survive the winter as adults mainly in the higher canopy of longan and eggs on *Melia azedarach* L., respectively (unpublished data, Hsu). However, the earlier nymphal instars of *S. formosana* prefer *Ficus fistulosa*, which is more likely found in humid valleys [31,32]. This might explain why in some drier areas rich in *E. decipiens* and *M. compressa*, we detected no egg masses of *S. formosana*. As for the food plants, the feeding behavior performed by adults was sighted on *E. decipiens*, *M. compressa*, and *T. glabrifolium*. In terms of nymphs, only the feeding on *H. heptaphyllum* has been observed because of a photograph taken from the front of a fifth-instar nymph. The mating and adult emergence have never been found probably due to the timing being in the night or early morning. It is not like the easy sighting of those behaviors in *P. candelaria* and *P. watanabei* on *T. sebifera* [2,6].

## 5. Conclusions

In the present study, we established that while the adults of *Saiva formosana* prefer *Elaeocarpus decipiens*, they may also be detected on *Magnolia compressa* and *Tetradium glabrifolium*. They tend to lay a few eggs per mass on a broad range of trees. This may be a good strategy for this lanternfly species to survive under the stress of its parasitic wasps, even though its eggs lack the protection of a wax cover. We suggest that egg masses, both hatched and unhatched, can be used for monitoring lanternflies as they are immobile, have a year-round presence, and are much easier to locate than nymphs or adults. In this study, we usually detected egg masses in the humid valley, probably due to the preference for *Ficus fistulosa* exhibited by early-instar nymphs. In addition, the older nymphal instars of *S. formosana* have a high rate of occurrence on *Heptapleurum heptaphyllum* after November. This overwintering strategy of this lanternfly species is similar to that of *Pyrops watanabei*. We may say, then, that the coexistence of *S. formosana* and *P. watanabei* in the same habitat is not unusual in Taiwan. Our results also indicated that the main host plants of *S. formosana* and *P. candelaria* are different. Therefore, the survival stress facing this lanternfly from invasion by *P. candelaria* is probably insignificant. Furthermore, we suggest that the prevalent egg parasitism of *S. formosana* may cast new light on a possible way to imply biological control agents for the invasive lanternfly, *P. candelaria*, in Taiwan. Finally, the identity of the parasitic wasp is an issue that must be addressed in future studies by specialists in the field of taxonomy.

## Figures and Tables

**Figure 1 insects-15-00841-f001:**
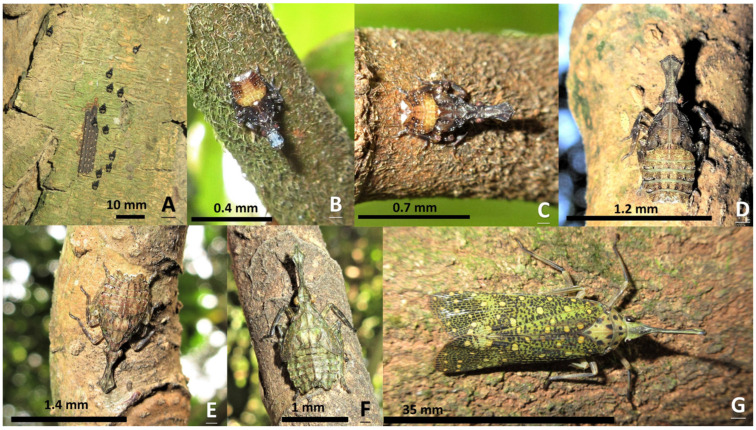
Different stages of *Saiva formosana*. (**A**) A batch of newly hatched nymphs beside an egg mass on *Acacia confusa*. (**B**) A first instar on *Ficus fistulosa*. (**C**) A second instar on *F. fistulosa*. (**D**) A third instar on *Heptapleurum heptaphyllum*. (**E**) A fourth instar on *H. heptaphyllum*. (**F**) A fifth instar on *H. heptaphyllum*. (**G**) An adult on *Elaeocarpus decipiens*.

**Figure 2 insects-15-00841-f002:**
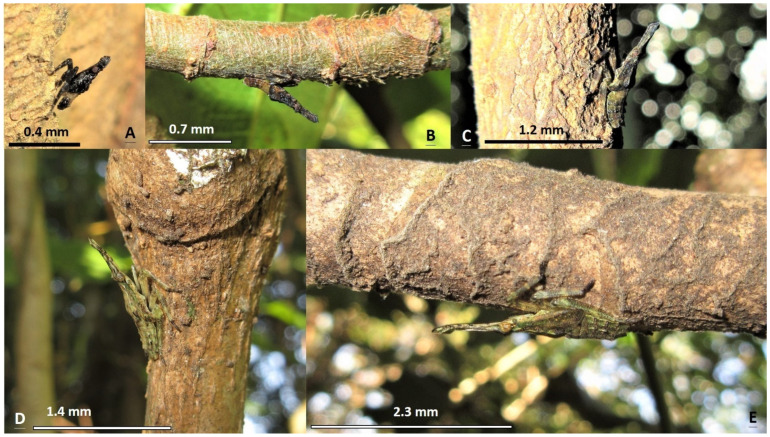
Side views of nymphs of *Saiva formosana*. (**A**) A first instar on *Acacia confusa*. (**B**) A second instar on *Ficus fistulosa*. (**C**) A third instar on *Machilus thunbergia*. (**D**) A fourth instar on *Heptapleurum heptaphyllum*. (**E**) A fifth instar on *H. heptaphyllum*.

**Figure 3 insects-15-00841-f003:**
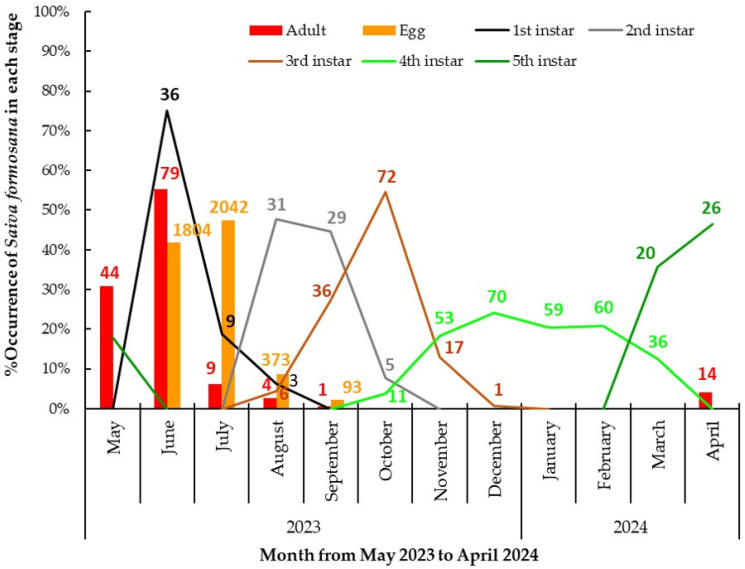
The percentage occurrence of *Saiva formosana* in each stage per month from 1 May 2023 to 30 April 2024. In terms of each stage, the sum of the 12 monthly percentage occurrences is 100% in one year. The numbers of individuals are labeled on the bars and curve lines.

**Figure 4 insects-15-00841-f004:**
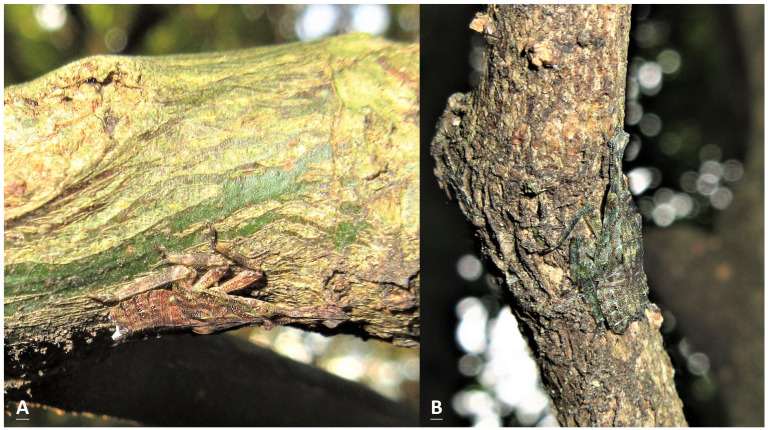
The whole body of a nymph in the fourth instar attached closely to the twigs of (**A**) *Citrus maxima* (19 December 2023), and (**B**) *Machilus thunbergia* (on 24 January 2024).

**Figure 5 insects-15-00841-f005:**
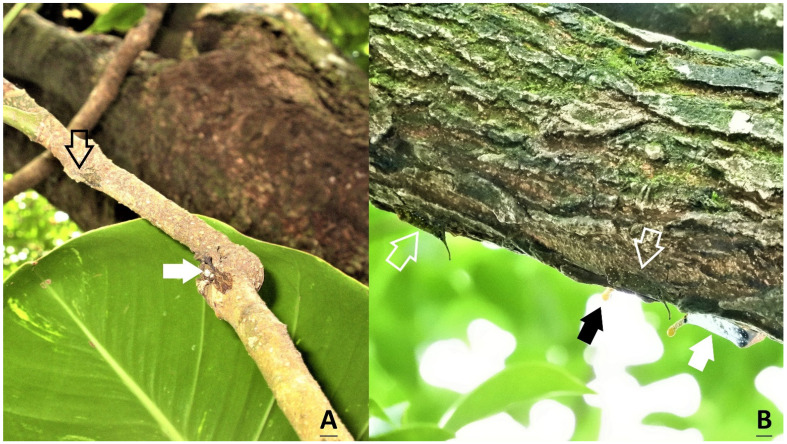
Coexistence of two species of lanternflies, *Saiva formosana* (hallow arrow) and *Pyrops watanabei* (solid arrow) on the same tree. (**A**) A fourth instar of *S. formosana* and a fourth instar of *P. watanabei* on *Heptapleurum heptaphyllum* (25 December 2023). (**B**) Two adults of *S. formosana* and two adults of *P. watanabei* on *Triadica sebifera* (14 June 2023).

**Figure 6 insects-15-00841-f006:**
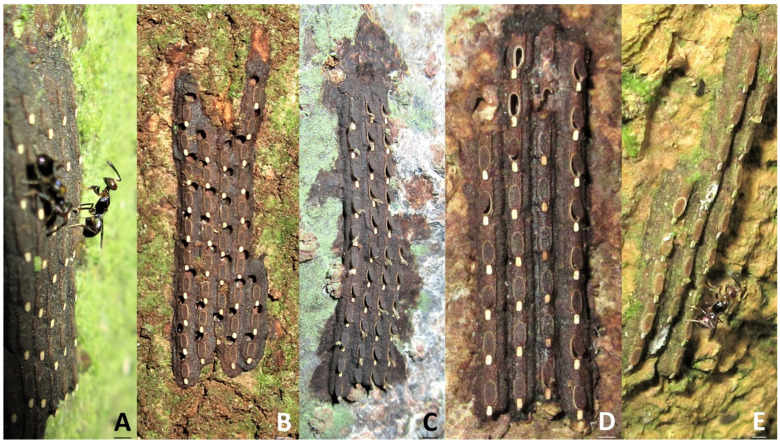
Egg masses of *Saiva formosana*. (**A**) An unhatched egg mass with two ovipositing parasitic wasps (*Anastatus* sp.). (**B**) An egg mass in which almost every egg has an emergence hole of a parasitic wasp. (**C**) A newly hatched egg mass with lids ajar and exuviae. (**D**) An egg mass in which only three eggs have lost their lids, while most eggs have retained theirs, and been re-covered. (**E**) An old, hatched egg mass with all the lids re-covered and a parasitic wasp. In contrast to the dark brown color of newly laid egg masses, this egg mass was light brown with apparent moss growth. It could be considered an egg mass laid prior to 2023.

**Figure 7 insects-15-00841-f007:**
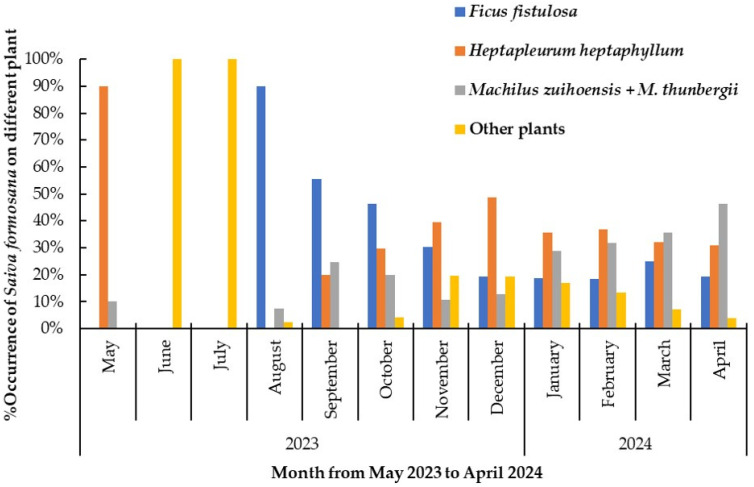
The percentage occurrence of nymphs of *Saiva formosana* on different host plants per month from 1 May 2023 to 30 April 2024. The percentage occurrence of this lanternfly is 100% in each month.

**Figure 8 insects-15-00841-f008:**
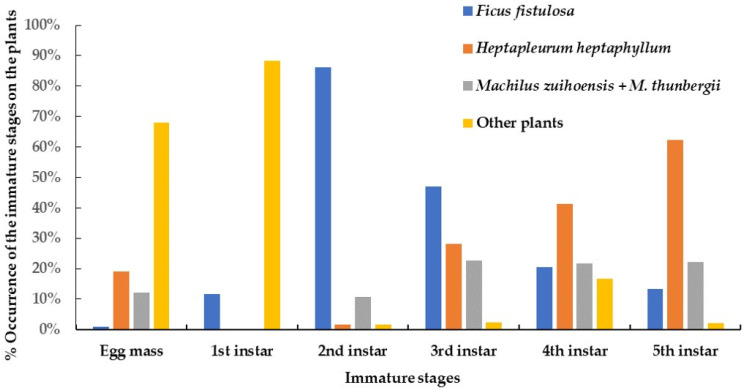
The shift of host range with different developmental immature stages from the egg mass to the nymph in the fifth instar of *Saiva formosana* recorded on different host plants from 1 May 2023 to 30 April 2024.

**Table 1 insects-15-00841-t001:** Investigation sites of the present study, with elevations and GPS coordinates. The stages of *Saiva formosana* Kato detected at the sites are also indicated.

Investigation Sites	Elevations	GPS Coordinates	Stages Detected
Erbazi Botanical Garden, Xindian, New Taipei City	285 m	24.936, 121.498	Adults, Eggs, and Nymphs
Miagaotai Peak Trail, Xinyi, Taipei City	260 m	25.015, 121.579	Adults, Eggs, and Nymphs
Zhanghu Hiking Trail, Wenshan, Taipei City	354 m	21.966, 121.579	Adults, Eggs, and Nymphs
Quantoumu Trail, Sanxing, Ilan County	212 m	24.658, 121.610	Adults, Eggs, and Nymphs
Tukuyue Trail, Nangang, Taipei City	272 m	25.026, 121.635	Adults and Eggs
Daijian Trail, Xizhi, New Taipei City	350 m	25.053, 121.665	Eggs and Nymphs
Mt. Shitou Trail, Xindian, New Taipei City	118 m	24.959, 121.544	Eggs
Dagemen Historical Trail, Shiding, New Taipei City	330 m	24.953, 121.667	Eggs
Caonan Benyan Tree, Wenshan, Taipei City	250 m	24.968, 121.608	Eggs
Paoma Historical Trail, Jiaoxi, Ilan County	322 m	24.849, 121.775	Eggs
Maokong Gondola Station, Wenshan, Taipei City	299 m	24.968, 121.589	Eggs
Houshanyue Trail, Wenshan, Taipei City	350 m	24.980, 121.599	Eggs
Mt. Shiqiliao Trail, Sanxia, New Taipei City	180 m	24.957, 121.476	Eggs

**Table 2 insects-15-00841-t002:** Numbers of adults of *Saiva formosana* on different plants, as recorded from 1 May to 22 September 2023 and from 20 to 25 April 2024.

Plant Species	Family	*n*	No. Adults
*Elaeocarpus decipiens*	Elaeocarpaceae	56	105
*Magnolia compressa*	Magnoliaceae	17	21
*Tetradium glabrifolium*	Rutaceae	8	10
*Triadica sebifera*	Euphorbiaceae	1	5
*Zanthoxylum ailanthoides*	Rutaceae	1	1
*Morus australis*	Moraceae	1	1
Sum	-	84	143

**Table 3 insects-15-00841-t003:** Numbers of egg masses or eggs of *Saiva formosana* on different plants, as recorded from 5 June to 9 September 2023.

Plant Species	Family	*n* ^a^	No. Egg Masses	No. Eggs	No. *Anastatus* sp. ^b^ on Egg Masses
*Heptapleurum heptaphyllum*	Araliaceae	20	22	791	0
*Magnolia compressa*	Magnoliaceae	16	17	647	3
*Machilus zuihoensis*	Lauraceae	9	9	327	0
*Mallotus paniculatus*	Euphorbiaceae	8	8	306	2
*Acacia confusa*	Fabaceae	4	7	281	1
*Acer serrulatum*	Sapindaceae	6	7	255	1
*Elaeocarpus decipiens*	Elaeocarpaceae	6	6	239	4
Dead tree ^c^	-	5	5	199	0
*Machilus thunbergii*	Lauraceae	5	5	175	1
*Turpinia formosana*	Staphyleaceae	4	4	117	0
*Wendlandia formosana*	Rubiaceae	2	2	86	1
*Quercus glauca*	Fagaceae	2	2	79	0
*Ficus ampelos*	Moraceae	2	2	62	2
*Ilex micrococca*	Aquifoliaceae	1	2	78	0
*Cleyera japonica*	Pentaphylacaceae	2	2	71	0
*Trema orientalis*	Cannabaceae	1	2	70	1
*Randia cochinchinensis*	Rubiaceae	1	1	49	0
*Ficus fistulosa*	Moraceae	1	1	34	0
*Zanthoxylum ailanthoides*	Rutaceae	1	1	47	0
*Ilex asprella*	Aquifoliaceae	1	1	43	0
*Hydrangea chinensis*	Hydrangeaceae	1	1	42	0
*Macaranga tanarius*	Euphorbiaceae	1	1	41	0
*Ilex ficoidea*	Aquifoliaceae	1	1	41	0
*Prunus persica*	Rosaceae	1	1	38	0
*Glochidion rubrum*	Phyllanthaceae	1	1	37	1
*Triadica cochinchinensis*	Euphorbiaceae	1	1	35	0
*Ardisia sieboldii*	Primulaceae	1	1	31	0
*Citrus maxima*	Rutaceae	1	1	27	0
*Diospyros eriantha*	Ebenaceae	1	1	23	0
*Daphniphyllum glaucescens*	Daphniphyllaceae	1	1	20	0
Sum	-	107	116	4291	17

^a^ only plants with unhatched egg masses were noted. ^b,c^ species were not identified.

**Table 4 insects-15-00841-t004:** Numbers of nymphs in different instars of *Saiva formosana* recorded on different plant species from 2 May 2023 to 25 April 2024.

Plant Species	Family	*n*	No. Nymphs					Total ^a^
			1st Instar	2nd Instar	3rd Instar	4th Instar	5th Instar	
*Ficus fistulosa*	Moraceae	165	6	56	62	64	12	200
*Heptapleurum heptaphyllum*	Araliaceae	176	0	1	37	129	22	189
*Machilus zuihoensis*	Lauraceae	47	0	0	0	50	17	67
*Machilus thunbergii*	Lauraceae	41	0	7	30	18	3	58
*Magnolia compressa*	Magnoliaceae	1	36 ^b^	0	0	0	0	36
*Callicarpa formosana*	Lamiaceae	22	0	0	0	30	1	31
*Ilex uraiensis*	Aquifoliaceae	8	0	0	3	7	0	10
*Citrus maxima*	Rutaceae	7	0	0	0	8	0	8
*Acacia confusa*	Fabaceae	1	6 ^b^	0	0	0	0	6
*Premna serratifolia*	Lamiaceae	5	0	0	0	5	1	6
*Turpinia formosana*	Staphyleaceae	2	3 ^b^	1	0	0	0	4
*Elaeocarpus decipiens*	Elaeocarpaceae	1	0	0	0	1	0	1
*Glochidion zeylanicum*	Phyllanthaceae	1	0	0	0	1	0	1
Sum	-	477	51	65	132	313	56	617

^a^ total number of nymphs of all instars on a plant species. ^b^ the newly hatched nymphs were found beside an egg mass on 20 June and 12 July 2023.

**Table 5 insects-15-00841-t005:** Rearing experiment for egg masses. Forty egg masses were reared in the laboratory to explore the impact of egg parasitism on the survival of eggs. Four situations are presented, with the numbers and chances of egg masses harvesting lanternfly nymphs only, parasitic wasps (*Anastatus* sp.) only, both, or none. “Win” denotes there is any nymph or wasp harvested from an egg mass. “Lose” denotes there is no nymph or wasp harvested from an egg mass.

No. Egg Masses	Win–Lose ^a^		Win–Win ^b^		Lose–Win ^c^		Lose–Lose ^d^	
	*n*	%	*n*	%	*n*	%	*n*	%
40	22	55.0	2	5.0	14	35.0	2	5.0

^a^ only the first-instar nymphs of *Saiva formosana* hatched from the egg mass. ^b^ both lanternfly nymphs and adult parasitic wasps were harvested. ^c^ only adult parasitic wasps emerged from the egg mass. ^d^ no insects were harvested.

## Data Availability

The original contributions presented in the study are included in the article and Appendix A, further inquiries could be directed to the corresponding author.

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
