# Peer review of "Host Plants for the Lanternfly *Saiva formosana* Kato, 1929 (Hemiptera, Fulgoridae) Endemic to Taiwan, and Parasitism of Its Eggs by Wasps"

_insects, 2024, doi:10.3390/insects15110841_

Round 1

Reviewer 1 Report

Comments and Suggestions for Authors This is an interesting paper providing a wealth of new primary data on the fulgorid species Saiva formosana, and it is certainly worth publishing as it offers novel insights. However, the discussion on waxes is somewhat too general and should be revised to be more precise (see further details below). The same applies to the host plant section, which is solid but would benefit from additional notes on the necessity to better distinguish between the types of host plants in planthoppers in the future.   Photos are important and necessary and should not be reduced in size. However some of them are not of top quality. If possible the authors should provide better ones - but this might be due to the low to the format sent for the review.
General comments:
- Each time a taxon (family, genus species) of an insect or a plant is reported for the first time in the text provide author and date of its original description (including in tables or provide separately as a supplement a complete list with full taxonomic names of plants and their family, all with author and dates) - Start presenting the taxonomic position of lanternflies, a vernacular name for some planthoppers of the family Fulgoridae in Hemiptera Fulgoromorpha  - the waxe issue L 309-339 : The authors seem to suggest that the waxes excreted and present on the bodies of larvae and adults are the same as those deposited by females to cover the eggs. This is likely inaccurate. Wax excretion occurs in all hemipterans, particularly in fulgoromorphs. The waxes are produced by tegumentary wax glands, which come in several types (lacquer, flocculent, solid, etc.) and excrete waxes of different compositions (Bourgoin T., 1986. DOI: 10.1080/21686351.1986.12278415). These secretions serve various functions: lubricant, anti-evaporation, protective shell formation (e.g., in scale insects) for liquid waxes, decoys against predation in many fulgoromorph larvae, and as a carrier for pheromones (e.g., Yoon et al., 2011 or Wang et al., 2018, in L. delicatula). In females Fulgoridae in particular, glands associated with the genital tract produce liquid secretions that agglutinate with other secretions produced by wax glands located in the last abdominal segments. By covering the eggs, these various secretions can provide a camouflage, protective, antifungal, and/or water-repellent function for the egg mass. In other words, there is no direct or necessary relationship between the waxes observed in larvae and those in adults, between those present in males and females, and finally between the waxes on the bodies of females and those deposited on the egg masses. This part needs some restructuring and rewriting. - the host plant issue. This is an interesting piece of the work. A conclusion which is not addressed in this paper would be to alert on the necessity to start differentiate in planthoppers between host plants such as: feeding plants, sheltering or resting plants and oviposting plants, … may be as is more usual in Psyllids Hemiptera where even successive primary and secondary plants (wintering host plants …) during the life of the species are separtely documented.   More precise comments: L.. 49. What means the ‘we’? who is concerned? It is no the same authors listed in the 4 papers to which the authors refer to. Should be probably rewritten to clearly identify the different teams having studied or state they all belong to the same lab or research unit ??? L. 51. provide taxonomic position of the host plant in addition of author and date L. 57. provide a ref. I suppose it is [6] L. 59. ref 7 does not address the genus Saiva in its phylogeny! it should be removed here. But a more complete and recent phylogeny addressing the two genera have been published since Urban & cryan 2009 that might be added: Bucher el al. 2023. L.59 replace “thus;" by “and”. Adults are easily differentiated but egg and larval stages more difficult. Rewrite this sentence. Fig3. A representation of the data with a curve overlaid on the histogram bars by egg, nymphal stages and adults would provide clearer insights. To be tested also in the other figures.
L. 298 delate one “we detected"

Comments on the Quality of English Language

I'm not a native English speaker. But the English looks fine.

Author Response

  1. Photos are important and necessary and should not be reduced in size. However some of them are not of top quality. If possible the authors should provide better ones - but this might be due to the low to the format sent for the review.

reply

Reply: Thank you for your reminding. We are afraid that the transformation to the on-line PDF format somewhat unclear. The photos is clear in the WORD format. We will also upload our PDF file in R1. We are also pleased to provide original photos if it is necessary.

  1. General comments: - Each time a taxon (family, genus species) of an insect or a plant is reported for the first time in the text provide author and date of its original description (including in tables or provide separately as a supplement a complete list with full taxonomic names of plants and their family, all with author and dates) – Start presenting the taxonomic position of lanternflies, a vernacular name for some planthoppers of the family Fulgoridae in Hemiptera Fulgoromorpha

Reply

We took your advice to revise our text. For the species names for plant species, we followed the usage of the botanic taxonomists. For example, we listed the species Magnolia compresssa Maxim. but not Magnolia compresssa Maxim., 1872. We arranged a table S1 as supplementary materials including more complete information on every plant species in Tables 2, 3, and 4.

  1. - the waxe issue L 309-339 : The authors seem to suggest that the waxes excreted and present on the bodies of larvae and adults are the same as those deposited by females to cover the eggs. This is likely inaccurate. Wax excretion occurs in all hemipterans, particularly in fulgoromorphs. The waxes are produced by tegumentary wax glands, which come in several types (lacquer, flocculent, solid, etc.) and excrete waxes of different compositions (Bourgoin T., 1986. DOI: 10.1080/21686351.1986.12278415). These secretions serve various functions: lubricant, anti-evaporation, protective shell formation (e.g., in scale insects) for liquid waxes, decoys against predation in many fulgoromorph larvae, and as a carrier for pheromones (e.g., Yoon et al., 2011 or Wang et al., 2018, in L. delicatula). In females Fulgoridae in particular, glands associated with the genital tract produce liquid secretions that agglutinate with other secretions produced by wax glands located in the last abdominal segments. By covering the eggs, these various secretions can provide a camouflage, protective, antifungal, and/or water-repellent function for the egg mass. In other words, there is no direct or necessary relationship between the waxes observed in larvae and those in adults, between those present in males and females, and finally between the waxes on the bodies of females and those deposited on the egg masses. This part needs some restructuring and rewriting.

Reply

Thank you very much for your good advice and providing useful literatures. We added a paragraph in the discussion section before we started to mention the wax on the different stages other than eggs. Three additional papers were cited for this additional paragraph. We also mentioned that wax covering may have functions other than escaping from predation or parasitism, such as camouflage, protection from desiccation, etc. More research needs to be done to address the issue of full functions performed by egg coverings.

  1. the host plant issue. This is an interesting piece of the work. A conclusion which is not addressed in this paper would be to alert on the necessity to start differentiate in planthoppers between host plants such as: feeding plants, sheltering or resting plants and oviposting plants, … may be as is more usual in Psyllids Hemiptera where even successive primary and secondary plants (wintering host plants …) during the life of the species are separtely documented.

Reply: In the results section, we added the sentences on the feeding behavior we sighted performing by adults. In cool season, the third- and fourth-instar nymphs usually sighted on the concave points on the twigs of Heptapleurum heptaphyllum. Therefore, we speculated that this is the plants serving as preferred shelters for overwintering. Other plants with low occurrence may only serve for resting. In discussion section, Lines 373-382 mentioned on the preference of egg-laying sites. Lines 399-405 discussed on the preference of trees to be the shelters for overwintering. The issues of food plants, and the preferred plants for mating and adult emergence also discussed in the last lines of the section. Lines 408-413.

  1. More precise comments: L.. 49. What means the ‘we’? who is concerned? It is no the same authors listed in the 4 papers to which the authors refer to. Should be probably rewritten to clearly identify the different teams having studied or state they all belong to the same lab or research unit ???

Reply

Thank you for your reminding. We or our” has been deleted in the introduction when mentioned about the past research or investigations. The passive voice has been used in the sentences. Lines 53-60.

  1. 51. provide taxonomic position of the host plant in addition of author and date

Reply: The sentence “The introduced host plant, Triadica sebifera (L.) Small, which belonged to Euphorbiaceae, is….” Has been added in the text. Lines 55-56. We arranged a table S1 as supplementary materials including more complete information on every plant species.

  1. 57. provide a ref. I suppose it is [6]

Reply: ref. [6] in the original version. Ref. [7] in this revised version. Lines 61-63.

  1. 59. ref 7 does not address the genus Saiva in its phylogeny! it should be removed here. But a more complete and recent phylogeny addressing the two genera have been published since Urban & cryan 2009 that might be added: Bucher el al. 2023.

Reply: Thank you for your information about Bubher et al. (2023). The reference has been removed and added Bubher et al. (2023) instead.

  1. 59 replace “thus;" by “and”. Adults are easily differentiated but egg and larval stages more difficult. Rewrite this sentence.

Reply: We have rewritten it in the text.

  1. A representation of the data with a curve overlaid on the histogram bars by egg, nymphal stages and adults would provide clearer insights. To be tested also in the other figures.

Reply: Figure 3 is too complicated. So it has been replaced by your suggestions to become a combination of a histogram overlaid by curve lines to provide clearer insight. However, other figures are keeping the same histogram bars.

11.L. 298 delate one “we detected"

Reply: We have revised the typo error “we detected”. Line 327.

Reviewer 2 Report

Comments and Suggestions for Authors

The lanternfly, Saiva formosana, is an endemic species of Taiwan, but knowledge of its life history is poorly documented. Elaeocarpus decipiens was reported to be the sole host plant for adults and fifth-instar nymphs of this species. This manuscript provides new and valuable biological information on this species based on a one-year intensive survey in northern Taiwan. The study found this lanternfly had a broad range of oviposition sites on 29 plant species of 21 families, nymphs preferred host plants belonged to 13 species of 12 families that differed from those preferred by adults, and five species of plants hosted adults. The results supported evidence to reveal that host plant preference shifted with time and developmental stage.

The description of the materials and methods, as well as the results and discussion, is detailed and clear, and the figures and tables are well-prepared. Overall, the manuscript is well-written, has high scientific merit, and does reach the publishing quality of Insects. I recommend accepting the manuscript after a minor revision. 

Some comments are listed below, which the authors may consider to improve the manuscript.

1.     Do not cite the Table or Figure number in parentheses at the end of the subheading in the Results section. See Lines 135, 154, 173, 210, 235, 251.

2.     Anastatus is a large genus of parasitic wasps belonging to the family Eupelmidae, and it is not easy to identify. It is recommended that experts be asked to identify the species status of the Anastatus sp. for further research in the future.

3.     Please see the marks and corrections in the attached PDF file for other comments.

Comments on the Quality of English Language

English writing is OK; minor editing of the English language is required.

Author Response

Thank you very much for your careful review on our manuscript. We tried to revise our manuscript based on your comments point by point.

1. Do not cite the Table or Figure number in parentheses at the end of the subheading in the Results section. See Lines 135, 154, 173, 210, 235, 251.

Reply: Thank you for your comments. We took all your suggestions and rewritten in the revised version. Lines 153, 170, 192, 230, 260, 276.

2. Anastatus is a large genus of parasitic wasps belonging to the family Eupelmidae, and it is not easy to identify. It is recommended that experts be asked to identify the species status of the Anastatus sp. for further research in the future.

Reply: Thank you for your comments. The taxonomy of the genus Anastatus in Taiwan is unclear and necessary to be revised. One of the authors of this manuscript cooperated with Dr. Gary Gibson on Taiwanese Anastatus based on DNA data and adult morphology. This Anastatus species in our study is probably an undescribed species based on the preliminary DNA data (COI, 16S) and adult morphological characters. The taxonomic work of this species will be carried out in the future.   

3. Please see the marks and corrections in the attached PDF file for other comments.

Reply: Thank you for your help with the careful review. We took almost all your comments and revised in the new version.

Reviewer 3 Report

Comments and Suggestions for Authors

Abstract: Not concise in the objective of the study.

Line 46 to 48 Reference an example of apparent competition and/or intraguild competition.

Introduction is written as a summary of previous studies with the same species.

Material and Methods

The structure of the section not concise. Results obtained should go to the results section.

Results

How do you differentiate the nymphs of the different species? Those are the results that should be included in the text.

Reference to the figure is not data

Figure 3. Use overlapped diagram bars for this figure. It is necessary a figure with the number of individuals. Only one figure is not sufficient to show the data.

Table 2. Were the adults actually feeding form those host plants?

Line 174 Have you checked the higher parts of the canopy? Specify in the material and methods section.

Parasitoid species are not identified.

Table 4. Were the different stages actually feeding form those host plants?

Figure 7 and 8 Same than figure 3

Table 5 difficult to interpret.

First parapgraph of discussion goes to results section.

Line 314 No statistic for that statement

Author Response

Thank you for your comments on our manuscript. We tried to revise our manuscript based on your comments point by point.

  1. Not concise in the objective of the study.

Reply: A sentence of the objective of this study has been added in the abstract. Lines 26-28. The number of words in the abstract has reduced from 210 to 194 words.

  1. Line 46 to 48 Reference an example of apparent competition and/or intraguild competition.

Reply: We have revised it in the content in lines 49-53.

  1. Introduction is written as a summary of previous studies with the same species.

Reply: Since Saiva formosana was first reported as a new species in Taiwan by Kato (1929), only another publication on the Lanternflies of Taiwan by Lin et al. (2023) has mentioned this species. Therefore, in the introduction we mentioned Saiva genus, and it closely related to Pyrops. The coexistence of three lanternflies, Pyrops candelaria (invasive to Taiwan), P. watanabei (native species), and S. formosana (endemic species) in the communal habitats are also introduced.

  1. Material and Methods, The structure of the section not concise. Results obtained should go to the results section.

Reply: We added some wording about using the powerful flashlight and digital camera with a 60x zoom for the checking of the higher parts of the canopy.

  1. How do you differentiate the nymphs of the different species? Those are the results that should be included in the text.

Reply: We added brief descriptions in the first paragraph of results section. For comparison, we also added the dorsal and lateral profile photographs of each instar of Pyrops candelaria and P. watanabei in Supplement. Lines 142-152.

  1. Reference to the figure is not data. Figure 3. Use overlapped diagram bars for this figure. It is necessary a figure with the number of individuals. Only one figure is not sufficient to show the data.

Reply: We changed Figure 3 to a combination chart with bars and curve lines, and the numbers of individuals are labelled on it.

  1. Table 2. Were the adults actually feeding form those host plants?

Reply: Feeding was often sighted on Elaeocarpus decipiens, Magnolia compressa, and Tetradium glabrifolium. Lines 181-182.

  1. Line 174 Have you checked the higher parts of the canopy? Specify in the material and methods section.

Reply: Yes! The 1000 lumens-flashlight can let us check the higher canopy and we use a digital camera with a 60x optical zoom to make sure the objects. It has been specified in material and methods section. For example, the adults and nymphs in Figure 5 were on the trees higher than 5 m. And, we added “Although egg masses can be found on higher parts of trunks or branches, most of …” in the results. Line 128.

  1. Parasitoid species are not identified.

Reply: Thank you for your comments. The taxonomy of the genus Anastatus in Taiwan is unclear and necessary to be revised. One of the authors of this manuscript cooperated with Dr. Gary Gibson on Taiwanese Anastatus based on DNA data and adult morphology. This Anastatus species in our study is probably an undescribed species based on the preliminary DNA data (COI, 16S) and adult morphological characters. The taxonomic work of this species will be carried out in the future. 

  1. Table 4. Were the different stages actually feeding form those host plants?

Reply: Lines 247-252. Feeding by nymphs was difficult to be observed in the field. However, high occurrences of nymphs were on F. fistulosa, H. Heptaphyllum, M. zuihoensis, and M. thunbergia. Therefore, these 4 species of plants might be food plants and winter shelters. Other plants may be served as the egg-laying or temporary resting sites in which the newly-hatched or fourth-instar nymphs can be detected.

  1. Figure 7 and 8 Same than figure 3

Reply: Figure 3 shows the percentage occurrence of S. formosana in each stage per month. This figure is nothing to do with host plants. So, Figure 3 is different with Figure 7. Moreover, Figure 8 is to deal with the relationship between the major host plants and different immature stages, and this is nothing to do with time. Therefore, Figure 8 is different with Figure 3. 

  1. Table 5 difficult to interpret.

Reply: Sorry about that! Sentences have been given to explain the four situations of rearing experiment for “egg masses” (Not eggs).

  1. First parapgraph of discussion goes to results section.

Reply: We have already added brief descriptions in the first paragraph of results section with supplement photographs. In the first paragraph of discussion, we discussed the results of S. formosana, with S. gemmata, Pyrops candelaria, and P. watanabei. We also mentioned on coexistence of three lanternflies. Therefore, please let us keep it in discussion section.